# Cytokine and Chemokine Retention Profile in COVID-19 Patients with Chronic Kidney Disease

**DOI:** 10.3390/toxins14100673

**Published:** 2022-09-28

**Authors:** Paola Ciceri, Valeria Bono, Lorenza Magagnoli, Matteo Sala, Antonella d’Arminio Monforte, Andrea Galassi, Alessandra Barassi, Giulia Marchetti, Mario Cozzolino

**Affiliations:** 1Department of Health Sciences, Renal Division, ASST Santi Paolo e Carlo, University of Milan, 20142 Milan, Italy; 2Department of Health Sciences, Clinic of Infectious Diseases, ASST Santi Paolo e Carlo, University of Milan, 20142 Milan, Italy; 3Laboratory of Clinical Biochemistry, Department of Health Sciences, ASST Santi Paolo e Carlo, University of Milan, 20142 Milan, Italy

**Keywords:** CKD, COVID-19, eGFR, mortality

## Abstract

Chronic kidney disease (CKD) patients are more susceptible to infections compared to the general population. SARS-CoV-2 virus pathology is characterized by a cytokine storm responsible for the systemic inflammation typical of the COVID-19 disease. Since CKD patients have a reduced renal clearance, we decided to investigate whether they accumulate harmful mediators during the COVID-19 disease. We conducted a retrospective study on 77 COVID-19 hospitalized subjects in the acute phase of the illness. Thirteen different cytokines were assessed in plasma collected upon hospitalization. The patients were divided into three groups according to their estimated glomerular filtration rate, eGFR < 30 (n = 23), 30 < eGFR < 60 (n = 33), eGFR > 60 mL/min (n = 21). We found that Tumor Necrosis Factor α and its receptors I and II, Interleukin-7, Leukemia Inhibitory Factor, FAS receptor, Chitinase 3-like I, and the Vascular Endothelial Growth Factor showed an increased accumulation that negatively correlate with eGFR. Moreover, non-survivor patients with an impaired kidney function have significantly more elevated levels of the same mediators. In conclusion, there is a tendency in COVID-19 ESRD patients to accumulate harmful cytokines. The accumulation seems to associate with mortality outcomes and may be due to reduced clearance but also to increased biosynthesis in most severe cases.

## 1. Introduction

Chronic Kidney Disease (CKD) is a life-threating disease with a great impact on the global health system. CKD and impaired kidney function resulted in 2017, according to ‘The Global Burden of Disease’ study, as the 12th cause of death world-wide with 4.6% of all-cause mortality ascribable to CKD and cardiovascular diseases due to CKD [1]. If cardiovascular disease is the first cause of death in these patients, and the second one is infections with infection-related hospitalizations that actively contribute to increase the mortality rate in this population. The higher susceptibility of CKD patients to infections is due to many factors such as uremia, dialysis access, the dialysis procedure per se, advanced age, malnutrition, hypoalbuminemia, burden of coexisting illnesses, immunosuppressive therapy, and vaccine hyporesponsiveness. For this reason, advanced CKD stages might be assumed as a state of acquired immunodeficiency [2].

Coronavirus pandemic started in 2019 and is still ongoing with more than six million deaths all over the world (World Health Organization, May 2022). The pathology is due to SARS-CoV-2 infection, a new RNA virus from the *coronoviridae* family, that causes the severe acute respiratory syndrome that characterizes the severe form of the disease. One of the more relevant events that can be triggered by the infection is a cytokine storm responsible for the systemic inflammation and prothrombotic state typical of the COVID-19 disease. The entire body is involved by the pathology and severe forms may evolve in death mainly for respiratory failure. In addition, kidneys are affected with development of renal damage that can lead to acute kidney injury (AKI). The etiology of AKI is probably multifactorial involving different processes. Directly the SARS-CoV-2 virus can infect kidney podocytes and proximal tubular cells causing tubular necrosis and protein leakage in Bowman’s capsule. Indirectly, the immune response alteration given by the virus, with the cytokine storm, lymphopenia, and macrophage activation, can contribute to AKI. In addition, lower oxygen delivery due to the acute respiratory syndrome can induce renal ischemic injury. Finally, endothelial dysfunction, hypercoagulability, rhabdomyolysis, and sepsis are other mechanisms potentially involved in renal damage [3].

Thus, the concern about COVID-19 disease in CKD patients is two-way since the infection might aggravate renal failure and the higher susceptibility to infections of these patients expose them to an increased risk of severe complications and higher mortality. Since renal insufficiency causes an increased difficulty in the clearance of many bioactive molecules, thus causing their accumulation, in this research we aimed to study the relationship between glomerular filtration and a wide range of cytokines, chemokines, and uremic toxins dysregulated by SARS-CoV-2 infection in CKD patients, in order to better understand the COVID-19 disease pathophysiology in this population and give a contribution to find strategies to better protect these vulnerable patients. 

## 2. Results

### 2.1. Clinical

The demographical and clinical characteristics of COVID-19 patients analyzed in our study are listed in Table 1. Seventy-seven hospitalized COVID-19 patients were enrolled: 48% were males and with a median age of 79 (IQR 70–86). The median time from symptoms onset was 4 days (IQR 2–8). In the overall population median eGFR was 48.4 mL/min/1.73 m^2^. Twenty (26%) patients reported history of CKD with four (5%) patients receiving maintenance hemodialysis. Twenty-three participants had eGFR < 30 mL/min/1.73 m^2^, 33 had 30 < eGFR < 60 mL/min/1.73 m^2^ and finally 21 had eGFR > 60 mL/min/1.73 m^2^. No differences were registered between groups in demographic and COVID-19-related parameters, i.e., symptoms, lung infiltrates, medical treatment/oxygen support, outcome. A higher proportion of patients with CKD was found in the eGFR<30 compared to 30 < eGFR < 60 and eGFR > 60 groups (70%, 12%, and 0%, respectively), whereas a higher proportion of patients with diabetes was found in the 30 < eGFR < 60 and eGFR < 30 compared to eGFR > 60 groups (42%, 39%, and 10%, respectively). A significant contraction of percentage of patients included in the age-adjusted Charlson score category 4 was described in the eGFR > 60 compared to eGFR < 30 and 30 < eGFR < 60 groups (43%, 48%, and 48%, respectively). The peripheral oxygen saturation level at admission was significantly higher in eGFR < 30 as well as 30 < eGFR < 60 (95, IQR 93–97, in both groups) compared to eGFR > 60 group (93, IQR 89–96). A higher neutrophil/lymphocyte ratio was observed in patients with eGFR < 30 compared to the other groups, whereas a higher proportion in serum level of alanine-aminotransferase ALT was found in the 30 < eGFR < 60 (24, IQR 18–33) compared to eGFR < 30 and eGFR > 60 groups (19, IQR 15–24 vs. 21, IQR 16–29, respectively). Serum creatinine was significantly higher in the eGFR < 30 (3.4, IQR 2.3–4.2) compared to 30 < eGFR < 60 and eGFR > 60 groups (1.2, IQR 1.2–1.4 vs. 0.7, IQR 0.6–0.9, respectively).

### 2.2. Characterization of COVID-19 Plasma with Respect of eGFR

Plasma levels of 13 different compounds were measured in COVID-19 patients. Molecules were chosen as biomarkers of inflammation, immunity response, tissue damage, and angiogenesis. TNF-α, TNFRI, and TNFRII showed an increased accumulation in patients with reduced renal function demonstrated by the negative and significant correlation between increased accumulation and the decline of eGFR (Figure 1A–C). Analyzing the TNF-α, TNFRI, and TNFRII levels with respect of eGFR and outcome, we found that non-survivor patients with eGFR < 30 had significant more elevated levels compared to non-survivor patients with 30 < eGFR < 60 and eGFR > 60 (Figure 1D–F). Moreover, non-survivor patients in the 30 < eGFR < 60 had a more elevated level of TNF-α TNFRI, and TNFRII compared to survivors (Figure 1D–F). Regarding IL-7, IL-18, IL-6 and IL-6R, only IL-7 showed an accumulation dependent on eGFR decrease and a significant more elevated levels in non-survivor patients with eGFR < 30 and 30 < eGFR < 60 compared with non-survivor patients with eGFR > 60 (Figure 2A,E). IL-18 and IL-6 showed increased levels in non-survivor patients with eGFR < 30 compared to non-survivor patients with eGFR > 60 (Figure 2F,G). IL-6 and IL-6R showed a trend with more elevated levels in non-survivor patients independently of eGFR values (Figure 2G,H). FAS, LIF, and YKL-40 had a significant negative correlation with eGFR reduction but not TRAIL (Figure 3A–D). FAS levels were significantly higher in non-survivor patients with eGFR < 30 decreasing significantly in non-survivor patients with more elevated eGFR (Figure 3E). The same trend was shown by LIF levels with a significant increase in non-survivor patients in the eGFR < 30 group (Figure 3F) and YKL-40 with the non-survivor patients in the eGFR > 60 that showed significant decreased levels compared to the other eGFR groups (Figure 3G). We next analyzed two acute phase proteins, AGP and PTX3, that did not show any accumulation dependent on eGFR decline (Figure 4A,B) but showed increased levels in the 30 < eGFR < 60 non-survivor group. Interestingly, PTX3 had more elevated levels in the non-survivors compared to survivor patients independently of eGFR value (Figure 4E). VEGF levels correlate with eGFR deterioration (Figure 4C) with a trend of being more elevated in non-survivor patients with a more pronounced eGFR decline (Figure 4F).

## 3. Discussion

COVID-19, in its life-threatening form, is characterized by severe acute respiratory syndrome and by acute organ failure caused by coronavirus 2 (SARS-CoV-2). The precise mechanisms and the immunopathogenesis of COVID-19 are still under study and debate, but mounting evidence suggests that virus-induced defective host immunity could be the cause of high mortality with a central role played by lymphopenia and monocytopenia. Among the different hypotheses to explain the decrease of the cells of the immune system, a role has been hypothesized for cell death [4]. In fact, probably the SARS-CoV-2 virus directly causes cell death by entering the immune cells and, indirectly, causes a massive increase in circulating cytokine and chemokines, the so-called ‘cytokine storm’, that is deleterious for immune cells probably participating in determining their death. Chronic kidney disease (CKD) is characterized by the decline in estimated glomerular filtration rate (eGFR) that leads to accumulation of uremic retention solutes, also defined as uremic toxins, that have a remarkable role in multiple organ system deterioration and general health in these patients. Therefore, in this study we aimed to characterize the retention profile of 13 among chemokines and cytokines chosen as biomarkers of inflammation, immunity response, tissue damage and angiogenesis in CKD COVID-19 patients and their association with kidney function, hypothesizing that the decreased renal clearance in CKD patients may induce an accumulation of mediators that because of their toxicity may aggravate the course of COVID-19 disease.

We dosed TNF-α as it is one of the master regulators of inflammation and a primary mediator of systemic response in sepsis and infections. TNF-α can both regulate cell apoptosis, necroptosis, and proliferation and stimulate other cytokine and chemokine biosynthesis. TNF-α coordinates the inflammatory response in the acute phase but to high TNF-α levels can suppress the immune system leading to adverse prognosis [5]. In COVID-19, it has been found that TNF-α levels are increased in severe compared to non-severe forms and correlate with disease severity, organ failure, and mortality [6]. One of the hypothesis of the mechanism of action in COVID-19 is that high TNF-α levels may aggravate lymphophenia by killing lymphocytes [7]. TNF-α has two receptors, TNFRI and TNFRII, that in their soluble form, sTNFR, are circulating and have been demonstrated to be involved in the immune cascade in inflammatory diseases such as septic shock. Moreover, sTNFR, and particularly TNFRII, were associated with increased risk of progression of diabetic kidney disease [8]. Even if circulating sTNFR act by binding TNF-α and by decreasing the number of receptor on the cellular membrane, the elevation of circulating levels has been associated with mortality during sepsis and with development of AKI in septic shock [9]. Moreover, it has been found that increased sTNFR levels are present in COVID-19 patients with the elevation associating with the severity of the disease and the prediction of AKI [10,11]. In our study, we found that there is an accumulation of both TNF-α and sTNFR inversely related to eGFR. These results may indicate that COVID-19 patients have an increased accumulation when a more impaired kidney function is present. Analyzing both TNF-α and sTNFR levels with respect of the mortality outcome, our data seems to confirm that as eGFR declines, non-survivor patients have increased levels compared to patients with more elevated eGFR values. Moreover, there is a clear trend in having increased both TNF-α and sTNFR levels in non-survivor patients compared to survivors independently of eGFR. 

We measured the levels of some cytokines, such as IL-7, IL-18, IL-6, and its receptor IL-6R. IL-7 is produced by multiple stromal cells and is involved in T cell development regulating their survival and homeostasis. For T cells, IL-7 is also anti-apoptotic and crucial for proliferation [12]. Administration of IL-7 increases both circulating and tissue lymphocytes, and for this effect it is currently under clinical trials for oncologic and infectious diseases [13]. In COVID-19, circulating IL-7 was found to be elevated [6] with levels that associate with disease severity [14]. In our population, we found that IL-7 levels inversely correlate with eGFR with an accumulation in patients with more impaired kidney function. Analyzing the mortality, IL-7 is increased in non-survivor COVID-19 patients with low eGFR, with a trend to be increased in non-survivors compared to survivor patients independently by renal function. IL-7 is one of the cytokines massively produced during the cytokine storm. The elevated levels have been associated with depletion of the T cell pool and may be due to a positive feedback response to lymphopenia. There are contradictory interpretations regarding the role of IL-7; in fact, evidence supports a beneficial role in COVID-19 as demonstrated in a small group of critically ill patients where the administration of IL-7 induced an increase in lymphocyte count without causing either lung damage or evident hyperinflammation [15]. A role for this cytokine as vaccine adjuvant has also been proposed [16]. Nevertheless, the augmentation of IL-7 above physiological levels may have a detrimental role disrupting the immunobalance [17]. Thus, the evidence in our population is that impaired renal clearance correlated with accumulation; whether this accumulation is detrimental or advantageous for the recovery of the disease needs deeper investigation. 

IL-18 is a pro-inflammatory cytokine belonging to the IL-1 family that is involved in the differentiation and activation of different T-cell population [18]. Overproduction of IL-18 may be detrimental since it can induce an exaggerated inflammatory response associated with an increase in morbidity and mortality. IL-18 has been described as part of the cytokine storm and a significative player in hemaphagocytic lymphohistiocytosis (HLH) and macrophage activation syndrome (MAS) [19]. Moreover, there is evidence that underlines as IL-18 is involved in injury induction in different organs such as lung, liver, and intestine and as its levels correlate with disease severity in sepsis, lupus erythematous and heart failure [20]. In COVID-19 patients, IL-18 is probably synthetized as part of the cytokine storm, and increased levels were found in non-survivor patients [21]. Moreover, IL-18 is more elevated in patients with most severe pneumonia and worse outcomes [22]. Our findings are in line with what reported in the literature, with a trend in more elevated levels in non-survivors compared with survivor patients. Moreover, even if there is not a correlation between IL-18 and kidney function decline we found a significant more elevated levels in non-survivor patients with impaired renal function (eGFR < 30) compared with patients with normal one. 

IL-6 is a pro-inflammatory cytokine that plays a central role in acute inflammation and is a driver of the cytokine storm. The IL-6 biological functions regarding the immune system are the promotion of T-cell population expansion and activation, B-cell differentiation, and the regulation of the acute phase response [23]. Many studies reported increased levels of IL-6 during COVID-19 with the more pronounced elevation associated with severity and adverse clinical outcomes [24]. Soluble IL-6 receptor (IL-6R) production is induced by IL-1beta and TNF-α and, differently from other cytokine circulating receptors, IL-6R is not inhibitory for IL-6 but, on the contrary, it activates IL-6 signalling. In our population, there is not a trend in accumulation of both IL-6 and IL-6R depending on the reduced renal clearance. Nevertheless, in line with other reported data, we found significant increased IL-6 levels in non-survivor patients with a trend for IL-6R. 

FAS is a death receptor for FAS-ligand (FASL), a pro-inflammatory cytokine that plays an important role in regulating apoptosis particularly in lymphocytes [25]. Physiologically, a soluble form of FAS (sFAS) exists that acts as a decoy receptor for FASL, thus decreasing sFAS-FASL signalling consequently inhibiting its pro-apoptotic effect. sFAS has also been proposed as a marker of inflammation and cardiovascular disease in uremia with increased levels in CKD patients [26]. It has been demonstrated that the FAS pathway is deeply involved in the pathogenesis of severe COVID-19 rather than susceptibility to the infection, and that elevated plasma FAS levels increase the risk of detrimental outcomes [27]. The hypothesis regarding the mechanism of action is that decreasing FASL signalling may result in impaired apoptosis of activated lymphocytes or of virus-infected cells. Our data demonstrate sFAS accumulation as kidney function declines in COVID-19 patients. Moreover, there are significant increased levels in non-survivor patients with reduced renal clearance compared with the ones with normal kidney function. 

The leukemia inhibitory factor (LIF) is a tissue factor belonging to the IL-6 cytokine family. At the kidney level, LIF regulates nephrogenesis protecting from oxidative stress and promoting tubular regeneration after acute renal failure [28]. At the pulmonary level, LIF is not produced in normal conditions but when alveolar macrophages, patrolling the blood-air barrier, encounter a virus, they release inflammatory cytokines as an alarm that triggers LIF production. LIF protects alveolar type I and II cells preventing scaring, fibrosis, and air niche to collapse [29]. LIF in animal models of pneumonia has been identified as a lung-protecting agent since it prevents severe disease development. In COVID-19, LIF levels are increased but no data on the associations with outcomes have been reported [30]. As LIF is a lung-protecting agent, the administration of recombinant LIF to protect the lung during COVID-19 has been proposed. The rationale is to prevent the severe forms and long term disease given its safety in already started phase I and II clinical trials [31]. In our population there is a significant accumulation of LIF when renal function declines and more increased levels in non-survivor patients with low eGFR compared to patients with improved kidney clearance. In light of LIF protecting pulmonary role, the increased levels in non-survivor patients might be interpreted as an index of the severity of the disease resulting in a more pronounced cytokine storm and tissutal pulmonary factor massive production. Nevertheless, impaired kidney function induced LIF accumulation and whether this might influence the disease outcome still need to be elucidated. 

3-Chitinase like 1 protein (YKL-40 in humans) is a tissutal factor produced in response to injury and cytokine stimuli that plays a major role in tissue damage, repair, remodelling, and in inflammation. YKL-40 levels are elevated in dialysis patients and associate with increased risk of progression of diabetic kidney disease [8]. YKL-40 is involved in various pulmonary chronic inflammatory diseases as asthma, virus-induced airway inflammation, and in interstitial lung disease where its levels associate with the severity of lung damage [32]. Moreover, it has a role in endothelial dysfunction regulating angiogenesis by an action on VEGF. In COVID-19, YKL-40 levels are increased and correlate with the disease severity. The hypothesized mechanism of action of YKL-40 during SARS-CoV-2 virus infection is the stimulation of the ACE2 receptor and viral spike protein priming proteases, demonstrated by the beneficial effects of YKL-40 and phosphorylation inhibitors [33]. Our data show a significant accumulation of YKL-40 with the reduction of eGFR with increased levels in non-survivor patients with more impaired renal function compared with the ones with mild renal insufficiency. 

Alpha 1 acid glycoprotein (AGP) is an acute phase protein belonging to the immunoglobulin family. It acts as an anti-inflammatory and immunomodulatory factor participating to endothelial permeability, leukocyte extravasation and platelet aggregation. AGP in response to infection, inflammation and tissue injury seems to have an anti-neutrophil and anti-complement role. Moreover, increased levels of AGP seem to participate to the recognition of microbes, and to increase blood flow at the site of injury [34]. Elevation of AGP has also been associated with severity in several inflammatory disease and with mortality in sepsis [35]. In COVID-19, AGP has been shown to be increased in the first days of the disease [36]. In our population, even if AGP plasma levels do not show any correlation with renal function there is a tendency to an elevation of AGP in non-survivor patients with more impaired kidney clearance. 

Pentraxin 3 (PTX3) is an acute phase protein and a key component of immune humoral immunity rapidly synthetized by different cell types in response to microbial infection, tissue damage and different soluble factors such as TNF-α and IL-1. For its actions, PTX3 can be considered a humoral pattern recognition molecule that provides defence against infection playing several function in tissue repair [37]. In COVID-19, PTX3 has been proposed as a biomarker able to predict mortality at 28 days [38,39]. Our data do not show any correlation between PTX3 levels and kidney function but, in line with other published data, PTX3 levels are significantly more elevated in non survivors compared with survivor patients. 

One of the characteristic features of SARS-CoV-2 infection evolving into critical forms is the involvement of vasculature determining thrombotic and microvascular complications due to endothelial injury and angiogenesis. While endotheliopathy is a key factor in COVID-19 associated with coagulopathy, angiogenesis, being an element of neovascularization, actively participates to fibrosis development typical of COVID-19 disease. We measured the levels of VEGF, a growth factor, marker of angiogenesis, and of endothelial activation. VEGF is a glycoprotein constitutively synthetized in the lung, involved in repair mechanisms, able to induce epithelial regeneration, and important for capillary leak. In COVID-19, VEGF levels are elevated and remain sustained also in patients with long term COVID-19 symptoms [40]. One hypothesis is that VEGF increases pulmonary vascular permeability because an ACE-2 receptor regulatory mechanism on VEGF is lost due to SARS-CoV-2 virus decreasing ACE-2 receptor expression [41]. In our population, we found that VEGF accumulation is dependent on renal function decline with increased levels in patients with a more impaired kidney function and a tendency to an elevation in non-survivors compared to survivor patients. 

In summary, in our population of CKD patients, there is a tendency of accumulation of cytokines and chemokines that correlated with the impairment of renal clearance. The accumulation often is significant in non-survivor patients with lower kidney function compared with patients with an improved renal function. Except for IL-7 and LIF that may not have harmful effects, all the other mediators seem to have a detrimental role in exacerbating COVID-19 disease and in general immune system impairment, inflammation, and tissue damage. Some of them are uremic toxins such as TNF-α, IL-6, IL-18, and VEGF as defined by the EUTox database (The European Uremic Toxins (EUTox) database. Available online at www.uremic-toxins.org). For other mediators such as sTNFR and YKL-40, independently of SARS-CoV-2 virus infection course, there is evidence of an association with kidney impairment due to renal pathology and thus the accumulation being potentially harmful for the kidney. 

All the cytokines and chemokines dosed in our population have a molecular weight between 15 and 50 KDa, thus being middle molecules (MM) accordingly with a recent classification [42]. Over the past few years, new dialytic membranes are available to better remove MM, the medium cut-off membranes with the technique of expanded hemodialysis [43]. 

In conclusion, there is a tendency for COVID-19 CKD patients to accumulate more harmful cytokines and chemokines, and this accumulation seems to associate with mortality outcomes. The higher levels may be due to reduced clearance but also to increased biosynthesis in most severe cases. Furthermore, advanced CKD stages might be assumed as a state of acquired immunodeficiency, impairing both innate and adaptive responses, Nonetheless, given the tendency to accumulate higher levels of bioactive molecules in CKD patients and the different dialytic methods available for the clinicians, a careful analysis regarding the dialysis procedure for each patient may improve COVID-19 course and general patient outcome and wellbeing. 

## 4. Materials and Methods

We consecutively enrolled hospitalized patients in the acute phase of COVID-19 at the Clinic of Infectious Diseases and Tropical Medicine, University of Milan, ASST Santi Paolo e Carlo, Italy, between March and September 2020. Patients were stratified according to estimated glomerular filtration rate (eGFR) assessed by CKD-EPI formula at hospital admission (eGFR < 30 mL/min; >30 but <60 mL/min, >60 mL/min). This study was approved by the Institutional Ethics Committee (Comitato Etico ASST Santi Paolo e Carlo, Milan, Italy 2 June 2020; 2020/ST/049, 2020/ST/049_BIS, 11/03/2020); written informed consent was obtained from participants. All research was performed in accordance with the Declaration of Helsinki.

### 4.1. Plasma Cytokine Quantification

Plasma samples were collected in 77 CKD patients upon hospitalization due to COVID-19 disease within the first 7 days of the disease. A broad range of cytokines and chemokines was assessed by Luminex technology (Austin, TX, USA): Interleukin-18 (IL-18), Interleukin-7 (IL-7), IL-6 receptor (IL-6R), Tumor Necrosis Factor α (TNF-α), Tumor Necrosis Factor α receptor I and II (TNFRI and TNFRII), Leukemia Inhibitory Factor (LIF), FAS receptor, Chitinase 3-like I (YKL-40), Pentraxin-3 (PTX3), and Vascular Endothelial Growth Factor (VEGF). Alpha 1 acid glycoprotein (AGP) and IL-6 were measured by ELISA (Human a1AGP ELISA kit cod EH4326, Gentaur, Kampenhout, Belgium; Human IL-6 Immunoassay cod HS600C, R&D Systems, Minneapolis, MN, USA) following the manufacturer’s instructions. 

### 4.2. Statistical Analysis

Demographic, clinical, and biochemical characteristics at baseline were stratified according to eGFR (<30 mL/min/1.73 m^2^, 30–60 mL/min/1.73 m^2^, >60 mL/min/1.73 m^2^). Categorical variables were reported as rates (%) and continuous variables were reported as median (interquartile range). Comparisons among groups were performed by Kruskal–Wallis Test. 

Serum levels of inflammatory biomarkers were plotted according to eGFR group and outcome (survivors vs. non-survivors). Wilcoxon signed rank test was used for pairwise comparisons among these categories. Linear correlation between inflammatory biomarkers and eGFR was assessed by Pearson correlation test. *p*-Value for significance was set at <0.05. Analysis was conducted by R version 4.1.1.

## Figures and Tables

**Figure 1 toxins-14-00673-f001:**
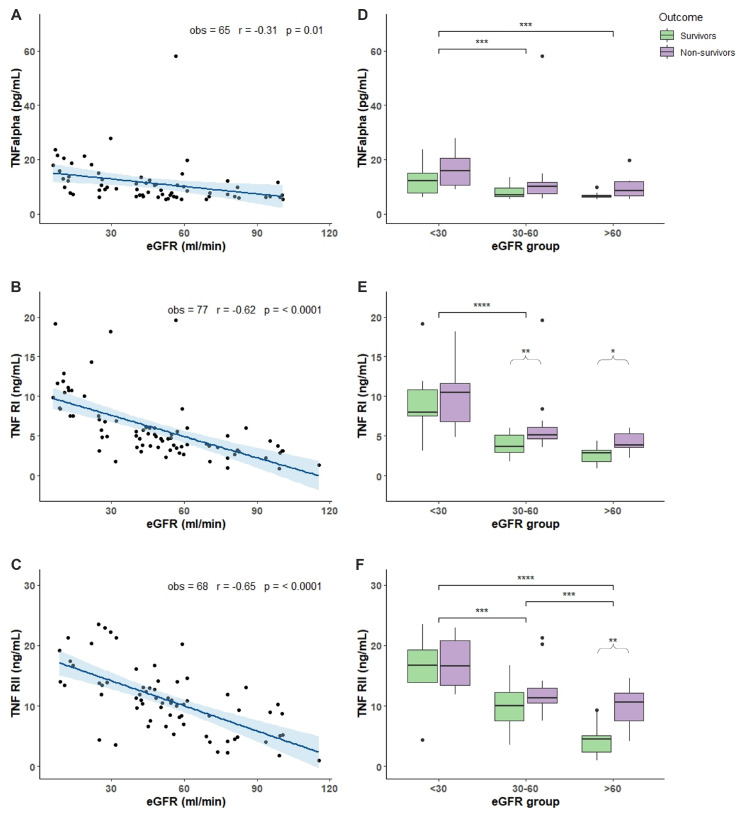
TNF-α and sTNF levels in relation to eGFR and mortality outcome. Plasma cytokine levels were measured in 77 patients hospitalized for COVID-19 in the acute phase of the illness within the first 7 days of COVID-19 disease. (**A**–**C**): correlation between TNF-α (**A**), TNFRI (**B**), and TNFRII (**C**) levels and eGFR values. (**D**–**F**): box plot representation of TNF-α (**D**), TNFRI (**E**), and TNFRII (**F**) levels classified accordingly with the outcome (survivors in green and non survivors in purple boxes) and three different ranges of eGFR values (<30, 30–60, and >60 mL/min). In each boxplot the thick line represent the median, the box represent the interquartile range, the whiskers represent the minimum and maximum score excluding outliers, and the dots represent outliers. * *p* < 0.05; ** *p* < 0.01; *** *p* < 0.001; **** *p* < 0.0001. Linear correlation between inflammatory biomarkers and eGFR was assessed by Pearson correlation test. Wilcoxon signed rank test was used for pairwise comparisons. *p*-Value for significance was set at <0.05. Analysis was conducted by R version 4.1.1.

**Figure 2 toxins-14-00673-f002:**
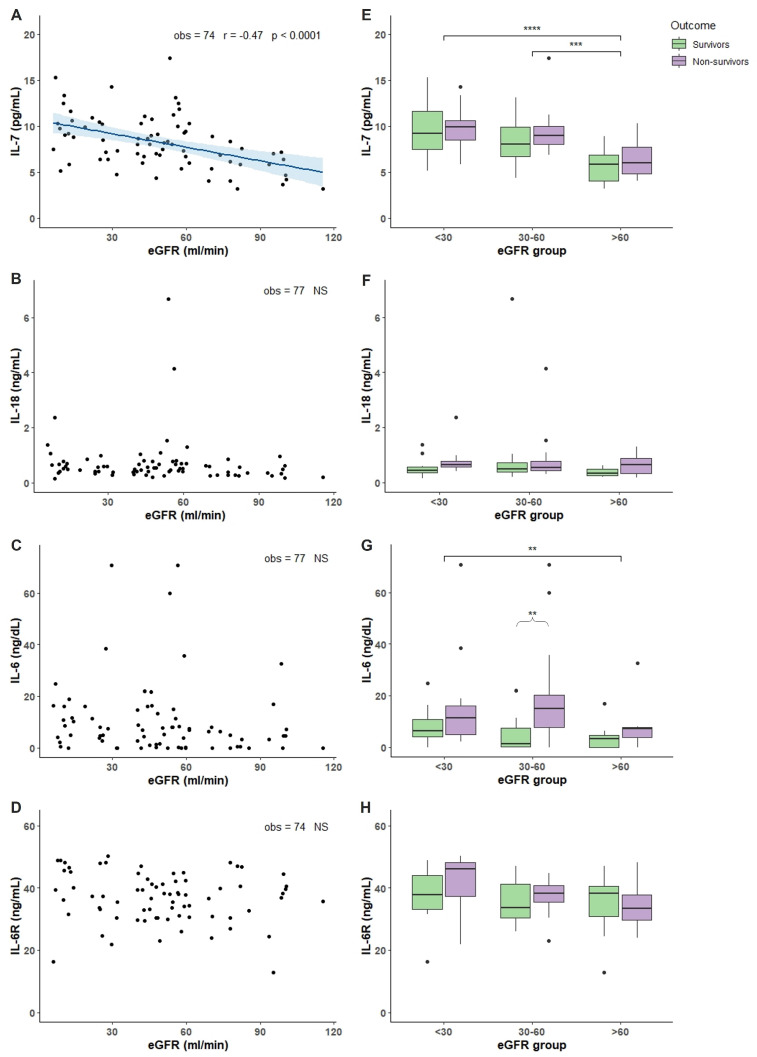
IL levels in relation to eGFR and mortality outcome. Plasma IL levels were measured in 77 patients hospitalized for COVID-19 in the acute phase of the illness within the first 7 days of COVID-19 disease. (**A**–**D**): correlation between IL-7 (**A**), IL-18 (**B**), IL-6 (**C**), and IL6R (**D**) levels and eGFR values. (**E**–**H**): box plot representation of IL-7 (**E**), IL-18 (**F**), IL-6 (**G**), and IL6R (**H**) levels classified accordingly with the outcome (survivors in green and non survivors in purple boxes) and 3 different ranges of eGFR values (<30, 30–60, and >60 mL/min). In each boxplot the thick line represent the median, the box represent the interquartile range, the whiskers represent the minimum and maximum score excluding outliers, and the dots represent outliers. ** *p* < 0.01; *** *p* < 0.001; **** *p* < 0.0001. Linear correlation between inflammatory biomarkers and eGFR was assessed by Pearson correlation test. Wilcoxon signed rank test was used for pairwise comparisons. *p*-Value for significance was set at <0.05. Analysis was conducted by R version 4.1.1.

**Figure 3 toxins-14-00673-f003:**
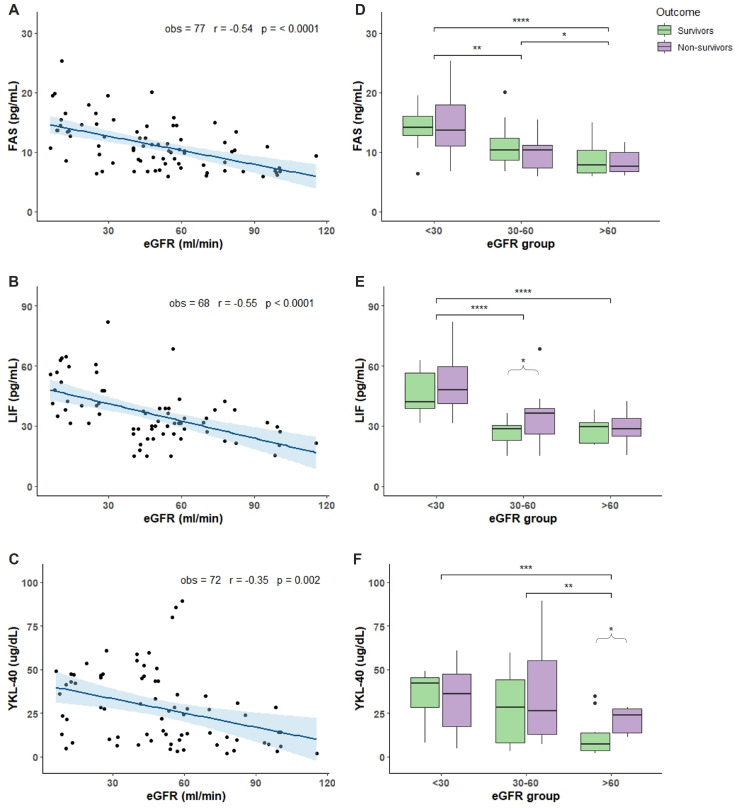
Chemokine levels in relation to eGFR and mortality outcome. Plasma chemokine levels were measured in 77 patients hospitalized for COVID-19 in the acute phase of the illness within the first 7 days of COVID-19 disease. (**A**–**C**): correlation between FAS (**A**), LIF (**B**), and YKL-40 (**C**) levels and eGFR values. (**D**–**F**): box plot representation of FAS (**D**), LIF (**E**), and YKL-40 (**F**) levels classified accordingly with the outcome (survivors in green and non survivors in purple boxes) and 3 different ranges of eGFR values (<30, 30–60, and >60 mL/min). In each boxplot the thick line represent the median, the box represent the interquartile range, the whiskers represent the minimum and maximum score excluding outliers, and the dots represent outliers. * *p* < 0.05; ** *p* < 0.01; *** *p* < 0.001; **** *p* < 0.0001. Linear correlation between inflammatory biomarkers and eGFR was assessed by Pearson correlation test. Wilcoxon signed rank test was used for pairwise comparisons. *p*-Value for significance was set at <0.05. Analysis was conducted by R version 4.1.1.

**Figure 4 toxins-14-00673-f004:**
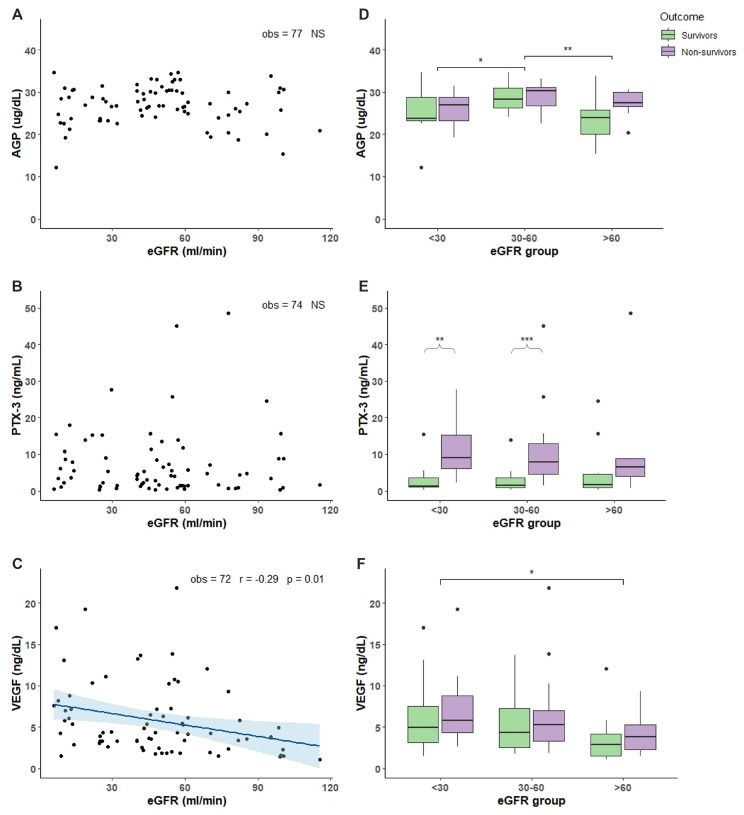
Acute phase protein and VEGF levels in relation to eGFR and mortality outcome. Plasma chemokine levels were measured in 77 patients hospitalized for COVID-19 in the acute phase of the illness within the first 7 days of COVID-19 disease. (**A**–**C**): correlation between AGP (**A**), PTX3 (**B**), and VEGF (**C**) levels and eGFR values. (**D**–**F**): box plot representation of AGP (**D**), PTX3 (**E**), and VEGF (**F**) levels classified accordingly with the outcome (survivors in green and non-survivors in purple boxes) and three different ranges of eGFR values (<30, 30–60, and >60 mL/min). In each boxplot the thick line represent the median, the box represent the interquartile range, the whiskers represent the minimum and maximum score excluding outliers, and the dots represent outliers. * *p* < 0.05; ** *p* < 0.01; *** *p* < 0.001. Linear correlation between inflammatory biomarkers and eGFR was assessed by Pearson correlation test. Wilcoxon signed rank test was used for pairwise comparisons. *p*-Value for significance was set at <0.05. Analysis was conducted by R version 4.1.1.

**Table 1 toxins-14-00673-t001:** Demographical and clinical characteristics of COVID-19 patients.

Demographical and Clinical Characteristics	Total 77	eGFR < 30 mL/min. 23 (30%)	eGFR 30–60 mL/min. 33 (43%)	eGFR > 60 mL/min. 21 (27%)	*p*-Value (group 0 Vs. 1 Vs. 2)
Sex, (n, %)					
M	37 (48%)	11 (48%)	20 (61%)	9 (43%)	
F	40 (52%)	12 (52%)	13 (39%)	12 (57%)	*p =* 0.3980
Age, (median, IQR)	79 (70–86)	79 (73–86)	79 (73–87)	75 (61–84)	*p* = 0.3513
Ethnicity, (n, %)					
White/Caucasian	72 (94%)	21 (91%)	32 (97%)	19 (90%)	
Latin American	3 (4%)	1 (4%)	0 (0%)	2 (10%)	
East Asian	1 (1%)		0 (0%)	0 (0%)	
Maghreb/Middle East	1 (1%)	0 (0%)	1 (3%)	0 (0%)	*p* = 0.3435
Comorbidities, (n, %)					
Hypertension	51 (66%)	19 (83%)	21 (64%)	11 (52%)	*p* = 0.0973
CVD	38 (49%)	14 (61%)	15 (45%)	9 (43%)	*p* = 0.4116
IMA	17 (22%)	6 (26%)	8 (24%)	3 (14%)	*p* = 0.5928
CHF	11 (14%)	5 (22%)	4 (12%)	2 (10%)	*p* = 0.4587
Arrhytmia	15 (19%)	6 (26%)	5 (15%)	4 (19%)	*p* = 0.6478
Valvulopathy	4 (5%)	1 (4%)	3 (9%)	0 (0%)	*p* = 0.3327
Cerebrovascular disease	8 (10%)	1 (4%)	4 (12%)	3 (14%)	*p* = 0.5090
Dementia	17 (22%)	6 (26%)	7 (21%)	4 (19%)	*p* = 0.8431
Chronic pulmonary disease	9 (12%)	2 (9%)	5 (15%)	2 (10%)	*p* = 0.7124
Cancer	9 (12%)	3 (13%)	6 (18%)	0 (0%)	*p* = 0.1244
CKD	20 (26%)	16 (70%)	4 (12%)	0 (0%)	*p* < 0.0001
Dialysis	4 (5%)	0 (0%)	0 (0%)	0 (0%)	-
Connettivopaties	2 (3%)	1 (23%)	1 (3%)	0 (0%)	*p* = 0.6495
Diabetes	25 (32%)	9 (39%)	14 (42%)	2 (10%)	*p* = 0.0302
Chronic liver disease	4 (5%)	1 (4%)	2 (6%)	0 (0%)	*p* = 0.3327
Vascular disease	16 (21%)	7 (30%)	7 (21%)	2 (10%)	*p* = 0.2319
Age adj. charlson score					
Category 1	6 (8%)	3 (13%)	0 (0%)	3 (14%)	*p* = 0.0861
Category 2	13 (17%)	2 (9%)	7 (21%)	4 (19%)	*p* = 0.4472
Category 3	22 (28%)	7 (30%)	10 (30%)	5 (24%)	*p* = 0.8517
Category 4	36 (47%)	11 (48%)	16 (48%)	9 (43%)	*p* = 0.0194
BMI, (median, IQR)	24.20 (22.92–29.07)	22.86 (20.43–23.74)	23.53 (22.15–29.30)	27.12 (24.34–29.02)	*p* = 0.2683
Symptoms at the admission. (n. %)					
Fever	53 (69%)	15 (65%)	22 (67%)	16 (76%)	*p* = 0.6900
Cough	26 (34%)	9 (39%)	11 (33%)	6 (29%)	*p* = 0.7588
Productive cough	3 (4%)	1 (4%)	1 (3%)	1 (5%)	*p* = 0.9415
Dyspnea	43 (56%)	15 (65%)	20 (61%)	8 (38%)	*p* = 0.1491
Fatigue	15 (19%)	4 (17%)	7 (21%)	4 (19%)	*p* = 0.9373
Abdominal pain	4 (5%)	2 (9%)	1 (3%)	1 (5%)	*p* = 0.6395
Nausea/vomiting	2 (3%)	0 (0%)	1 (3%)	1 (5%)	*p* = 0.5985
Diarrhoea	3 (4%)	1 (4%)	2 (6%)	0 (0%)	*p* = 0.5281
Chest pain	2 (3%)	0 (0%)	2 (6%)	0 (0%)	*p* = 0.2544
Syncope	1 (1%)	1 (4%)	0 (0%)	0 (0%)	*p* = 0.3044
Arthromyalgia	2 (3%)	0 (0%)	2 (6%)	0 (0%)	*p* = 0.2544
Anosmia/dysgeusia	3 (4%)	1 (4%)	1 (3%)	1 (5%)	*p* = 0.9415
Duration of symptoms before the hospitalization. days, (median, IQR)	4 (2–8)	4 (2–6)	5 (3–9)	5 (2–8)	*p* = 0.5657
Radiological pulmonary infiltrates upon admission, (n, %)	67 (87%)	20 (87%)	30 (91%)	17 (81%)	*p* = 0.5695
Respiratory setting upon admission, (n, %)					
Room air	60 (78%)	16 (70%)	29 (88%)	15 (71%)	
O2-therapy	17 (22%)	7 (30%)	4 (12%)	6 (29%)	*p* = 0.1873
Respiratory parameters upon admission, (median, IQR)					
pO2	72 (62–88)	63 (70–78)	70 (63–78)	71 (61–87)	*p* = 0.5435
pO2/FiO2	251 (307–357)	286 (305–351)	205 (286–351)	296 (209–343)	*p* = 0.2812
SpO2	96 (91–97)	95 (93–97)	95 (93–97)	93 (89–96)	*p* = 0.0140
Blood examinations upon admission. (median. IRQ)					
Hemoglobin, g/dL	12.0 (11.0–13.30)	11.6 (11.0–12.5)	12.0 (11.0–13.5)	12.5 (11.7–13.4)	*p* = 0.2495
WBC count, 10^3^/μL	7.00 (5.42–9.71)	7.27 (5.64–10.19)	7.33 (5.83–9.71)	6.45 (5.26–7.66)	*p* = 0.3568
Neutrophils, 10^3^/μL	4.93 (3.93–7.51)	5.15 (4.03–8.45)	5.82 (4.10–7.57)	4.13 (3.43–5.94)	*p* = 0.1097
Lymphocytes, 10^3^/μL	1.04 (0.64–1.34)	1.01 (0.64–1.21)	0.83 (0.63–1.45)	1.10 (0.73–1.59)	*p* = 0.3656
NL ratio	5.16 (3.12–9.81)	6.73 (3.48–12.75)	6.14 (3.77–11.13)	3.66 (2.60–5.29)	*p* = 0.0454
Monocytes, 10^3^/μL	0.54 (0.34–0.77)	0.53 (0.35–0.89)	0.57 (0.34–0.89)	0.49 (0.27–0.65)	*p* = 0.4117
Platelets 10^3^/μL	204 (162–304)	177 (145–259)	206 (167–312)	230 (176–293)	*p* = 0.4868
C-reactive protein, mg/L	68.9 (27.3–99.1)	75.8 (32.7–98.1)	67.6 (27.8–132.3)	60.0 (21.4–80.2)	*p* = 0.4752
LDH, U/L	288 (211–390)	307 (226–429)	279 (209–398)	293 (210–374)	*p* = 0.8047
Creatine-P-kinase, U/L	92 (45–189)	97 (44–175)	126 (67–203)	61 (39–181)	*p* = 0.2437
D-dimer, ng/mL	580 (310–1448)	980 (487–3189)	600 (268–899)	494 (260–908)	*p* = 0.1572
ALT, U/L	21 (16–29)	19 (15–24)	24 (18–34)	21 (16–29)	*p* = 0.0404
AST, U/L	37 (28–49)	31 (23–47)	39 (30–51)	34 (31–46)	*p* = 0.1101
Creatinin, mg/dL	1.3 (1.0–2.2)	3.4 (2.3–4.2)	1.2 (1.2–1.4)	0.7 (0.6–0.9)	*p* < 0.0001
Procalcitonin, ng/mL	0.13 (0.07–1.21)	0.26 (0.13–5.00)	0.11 (0.07–0.47)	0.09 (0.04–0.56)	*p* = 0.0889
Ferritin, ng/mL	436 (231–829)	401 (231–865)	478 (179–827)	436 (310–567)	*p* = 0.9435
eGFR	48.4 (26.2–61.3)	12.3 (8.7–24.9)	49.1 (43.0–55.7)	82.0 (73.6–98.4)	*p* < 0.0001
Medical therapy, (n, %)					
Lopinavir/darunavir	10 (13%)	3 (13%)	2 (6%)	5 (24%)	*p* = 0.1672
Hydroxychloroquine	57 (74%)	16 (70%)	26 (79%)	15 (71%)	*p* = 0.7044
Steroids	17 (22%)	8 (35%)	5 (15%)	4 (19%)	*p* = 0.2029
Heparin	58 (75%)	16 (70%)	29 (88%)	13 (62%)	*p* = 0.0727
Biological drug	10 (13%)	5 (22%)	4 (12%)	1 (5%)	*p* = 0.2419
Maximum respiratory support, (n, %)					
Room air	10 (13%)	1 (4%)	4 (12%)	5 (24%)	*p* = 0.1559
O_2_-therapy	32 (41%)	8 (35%)	16 (49%)	8 (38%)	*p* = 0.5515
C-pap	26 (34%)	11 (48%)	9 (27%)	6 (29%)	*p* = 0.2812
NIMV	7 (9%)	2 (9%)	4 (12%)	1 (5%)	*p* = 0.6547
Oro-tracheal intubation	2 (3%)	1 (4%)	0 (0%)	1 (5%)	-
Outcome, (n, %)					
Discharge	42 (55%)	8 (43%)	19 (57%)	13 (62%)	
Death	35 (45%)	13 (57%)	14 (43%)	8 (38%)	*p* = 0.4238
Time from symptoms onset to outcome, days, (median, IQR)	18 (11–35)	17 (10–50)	18 (12–35)	19 (14–26)	*p* = 0.9860

Clinical, demographic and biohumoral characteristics upon admission. All patients, eGFR < 30 group, 30 < eGFR < 60 group, and eGFR > 60 group. (IQR: interquartile range; CVD: cardio-vascular disease; MI: myocardial infarction; CHF: congestive heart failure; CKD: chronic kidney disease; BMI: body mass index; NL: neutrophils-lymphocytes; CRP: C-reactive protein; LDH: lactate dehydrogenase; CPK: creatin-phospho-kinase; ALT: alanine aminotransferase; AST: aspartate aminotransferase; eGFR: estimated glomerular filtration rate calculated by CDK-EPI formula). Continuous variables expressed as median, IQR. Categorical variables expressed as number, %.

## Data Availability

Renal Division ASST Santi Paolo and Carlo, Milan, Italy.

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
