# Peer review of "Cytokine and Chemokine Retention Profile in COVID-19 Patients with Chronic Kidney Disease"

_toxins, 2022, doi:10.3390/toxins14100673_

Round 1
Reviewer 1 Report
The authors well deacribe cytokine and chemokine retention profile in Covid-19 patients with CKD. The study, with widely shareable content, is methodologically correct and the results are well described.
I would like to suggest only few formal adjustments:
- In the title, “CKD” would have to be replaced with the extension “chronic kidney disease”;
- The word “Covid-19”, in the title and in the text, would have to be repored with capital letters (COVID-19”);
- To be remove the duble “.” at line 374
Author Response
Letter to Editor & Referees
Reviewer #1:
The authors well describe cytokine and chemokine retention profile in Covid-19 patients with CKD. The study, with widely shareable content, is methodologically correct and the results are well described.
I would like to suggest only few formal adjustments:
- In the title, “CKD” would have to be replaced with the extension “chronic kidney disease”;
- The word “Covid-19”, in the title and in the text, would have to be reported with capital letters (COVID-19”);
- To be remove the double “.” at line 374
Answer. We thank the Reviewer for her/his revision of our manuscript. We have changed the title and reported CVOID-19 in capital letters. Also, we removed double “.” At line 374
Reviewer #2:
This clinical research paper contains a clear description of cytokines and chemokines in relation to estimated glomerular filtration rate.
Answer. We thank the Reviewer for her/his revision of our manuscript.
- Do not use a non-standard abbreviation in the title. CKD will be known to many but not all. Write ‘chronic kidney disease’ in the title.
A: We have changed the title
- The abstract contains a whole series of abbreviations that are not explained. Simply remove the first list and state the number of cytokines and chemokines that were determined. For the chemokines and cytokines that were different, provide an explanation for the abbreviation at first use. All non-standard abbreviations must be explained at first use.
A: We have corrected the abstract following the suggestion
- Differences in levels do not obligatory imply differences in clearance. The immune system as such is distinct in patients with chronic kidney disease. This is also clear in relation to TNF-alpha levels. The relation with eGFR is weak. Correlation does not imply causation. Only very strong correlations strongly suggest a causal relationship.
A: We have modified the discussion inserting the sentence “In our study, we found that there is an accumulation of both TNF-α and sTNFR inversely related to eGFR. These results may indicate that COVID-19 patients have an increased accumulation when a more impaired kidney function is present”.
- Chronic kidney disease affects both innate and adaptive responses. Please discuss immune dysfunction in chronic kidney disease. In general, these patients are immunocompromised. It is not simply a question of clearance.
A: We have modified the conclusion, adding this sentence: Furthermore, advanced CKD stages might be assumed as a state of acquired immunodeficiency, impairing both innate and adaptive responses. Nonetheless, given the tendency”.
- TNF-α contains a hyphen.
A: We have corrected it
Reviewer 2 Report
This clinical research paper contains a clear description of cytokines and chemokines in relation to estimated glomerular filtration rate. 1. Do not use a non-standard abbreviation in the title. CKD will be known to many but not all. Write ‘chronic kidney disease’ in the title. 2. The abstract contains a whole series of abbreviations that are not explained. Simply remove the first list and state the number of cytokines and chemokines that were determined. For the chemokines and cytokines that were different, provide an explanation for the abbreviation at first use. All non-standard abbreviations must be explained at first use. 3. Differences in levels do not obligatory imply differences in clearance. The immune system as such is distinct in patients with chronid kidney disease. This is also clear in relation to TNF-alpha levels. The relation with eGFR is weak. Correlation does not imply causation. Only very strong correlations strongly suggest a causal relationship. 4. Chronic kidney disease affects both innate and adaptive responses. Please discuss immune dysfunction in chronic kidney disease. In general, these patients are immunocompromised. It is not simply a question of clearance. 5. TNF-α contains a hyphen.
Author Response

(The authors gave the same response as above.)
